# Left Heart Disease Phenotype in Elderly Patients with Pulmonary Arterial Hypertension: Insights from the Italian PATRIARCA Registry

**DOI:** 10.3390/jcm11237136

**Published:** 2022-11-30

**Authors:** Matteo Toma, Roberta Miceli, Edoardo Bonsante, Davide Colombo, Marco Confalonieri, Andrea Garascia, Stefano Ghio, Mariangela Lattanzio, Carlo Maria Lombardi, Giuseppe Paciocco, Cristina Piccinino, Irene Rota, Caterina Santolamazza, Laura Scelsi, Piermario Scuri, Davide Stolfo, Antonella Vincenzi, Lorenzo Volpiano, Marco Vicenzi, Pietro Ameri

**Affiliations:** 1Cardiovascular Disease Unit, Cardiac, Vascular, and Thoracic Department, IRCCS Ospedale Policlinico San Martino, Largo Rosanna Benzi 10, 16132 Genova, Italy; 2Cardiovascular Disease Unit, Ente Ospedaliero Ospedali Galliera, Mura delle Cappuccine 14, 16128 Genova, Italy; 3Cardiology Unit, Ospedale di Bolzano, 39100 Bolzano, Italy; 4Cardiology Unit, Fondazione IRCCS Policlinico San Matteo Pavia, 27100 Pavia, Italy; 5Department of Pulmonology, University Hospital of Cattinara, Strada di Fiume 447, 34149 Trieste, Italy; 6De Gasperis Cardio Center, Niguarda Hospital, 20162 Milano, Italy; 7Cardiology Unit, Department of Heart and Vessels, Ospedale di Circolo & Fondazione Macchi, 21100 Varese, Italy; 8Cardiology Unit, ASST Spedali Civili di Brescia, 25123 Brescia, Italy; 9Department of Medical and Surgical Specialties, Radiological Sciences and Public Health University of Brescia, 25123 Brescia, Italy; 10Pulmonary Unit, Cardio-Thoracic and Vascular Department, Milano Bicocca University, San Gerardo Hospital, 20126 Milano, Italy; 11Division of Cardiology, “Maggiore della Carità” Hospital, Corso Mazzini 18, 28100 Novara, Italy; 12Cardiovascular Disease Unit, Internal Medicine Department, Fondazione IRCCS Ca’ Granda Ospedale Maggiore Policlinico, 20122 Milano, Italy; 13Cardiology Unit, Cardiovascular Department, ASST- Ospedale Papa Giovanni XXIII, 24127 Bergamo, Italy; 14Cardiothoracovascular Department, Azienda Sanitaria Universitaria Giuliano Isontina and University Hospital of Trieste, 34129 Trieste, Italy; 15Division of Cardiology, Department of Medicine, Karolinska Institutet, 17177 Stockholm, Sweden; 16Department of Cardiology, San Gerardo Hospital, 20900 Monza, Italy; 17Department of Clinical Sciences and Community Health, Università degli Studi di Milano, 20122 Milano, Italy; 18Department of Internal Medicine, University of Genova, Viale Benedetto XV, 6, 16132 Genova, Italy

**Keywords:** pulmonary hypertension, elderly, left heart disease, comorbidity, vasodilator

## Abstract

Pulmonary arterial hypertension (PAH) in the elderly is often associated with left heart disease (LHD), prompting concerns about the use of pulmonary vasodilators. The PATRIARCA registry enrolled ≥70 year-old PAH or chronic thromboembolic pulmonary hypertension (CTEPH) patients at 11 Italian centers from 1 December 2019 through 15 September 2022. After excluding those with CTEPH, post-capillary PH at the diagnostic right heart catheterization (RHC), and/or incomplete data, 23 (33%) of a total of 69 subjects met the criteria proposed in the AMBITION trial to suspect LHD. Diabetes [9 (39%) vs. 6 (13%), *p* = 0.01] and chronic kidney disease [14 (61%) vs. 12 (26%), *p* = 0.003] were more common, and the last RHC pulmonary artery wedge pressure [14 ± 5 vs. 10 ± 3 mmHg, *p* < 0.001] was higher and pulmonary vascular resistance [5.56 ± 3.31 vs. 8.30 ± 4.80, *p* = 0.02] was lower in LHD than non-LHD patients. However, PAH therapy was similar, with 13 (57%) and 23 (50%) subjects, respectively, taking two oral drugs. PAH medication patterns remained comparable between LHD and non-LHD patients also when the former [37, 54%] were identified by atrial fibrillation and echocardiographic features of LHD, in addition to the AMBITION criteria. In this real-world snapshot, elderly PAH patients were treated with pulmonary vasodilators, including combinations, despite a remarkable prevalence of a LHD phenotype.

## 1. Introduction

Pulmonary hypertension (PH) is classified in five groups according to hemodynamic profile and pathological findings [1]. Group 1 PH, or pulmonary arterial hypertension (PAH), has historically been described in young patients. For instance, in the 1980s, within US National Institutes of Health (NIH) registry of PAH, the mean age was 36 ± 15 years [2]. By contrast, the elderly population mostly shows a post-capillary PH profile associated with left heart disease (LHD, group 2 PH), pre-capillary PH, but in the presence of severe lung disease (group 3 PH), or chronic thromboembolic pulmonary hypertension (CTEPH) [3,4].

Over the past years, several authors have reported that the age at PAH diagnosis is rising. In the Swiss Pulmonary Hypertension registry, which has enrolled PAH patients since 1998, the mean age has increased from 53 ± 16 years between 2000 and 2004 to 60 ± 15 years in the period between 2009 and 2012 [5]. A sizable proportion of subjects diagnosed with PAH in the elderly was also found in the French National registry, the US Registry to evaluate early and long-term pulmonary arterial disease management (REVEAL), and the European multicenter Comparative, Prospective Registry of Newly Initiated Therapies for Pulmonary Hypertension (COMPERA) [6,7,8]. 

Compared to younger PAH patients, older ones more often have risk factors for LHD, such as coronary artery disease (CAD), systemic hypertension, diabetes, and chronic kidney disease (CKD), or established LHD [9,10]. This may, in turn, affect the efficacy and safety of pulmonary vasodilators. A pre-specified analysis of the Ambrisentan and Tadalafil in Patients with Pulmonary Arterial Hypertension (AMBITION) trial compared patients with and without a LHD phenotype, according to clinical and hemodynamic parameters, as defined by a protocol amendment [11]. Compared to the trial participants without a LHD profile, those with a LHD phenotype were older (62.1 ± 10.2 vs. 54.4 ± 14.6 years) and had lower six-minute walk distance (6MWD) (330.5 vs. 363.7 m), mean pulmonary artery pressure (mPAP, 42.2 ± 12.4 vs. 48.7 ± 12.5 mmHg), and pulmonary vascular resistance (PVR, 512.1 ± 293.2 vs. 824.9 ± 402.1 dyne*sec/cm^5^). Furthermore, they benefited less from PAH therapy and discontinued it more frequently [11].

We evaluated the presence of a LHD phenotype and the pattern of pulmonary vasodilator prescription in a cohort of elderly patients with PAH enrolled in the multicenter PATRIARCA registry.

## 2. Materials and Methods

PATRIARCA (Registro dell’iPertensione ArTeriosa polmonaRe e ipertensIone polmonAre cRonica tromboemboliCa nell’Anziano) is a registry conducted at 11 centers in Northern Italy assessing the characteristics of PAH and inoperable, persistent, or relapsing CTEPH in the elderly. Following ≥20 PAH/CTEPH patients was required to join the study as an indicator of sufficient expertise in these diseases.

The registry consists of 2 phases: a cross-sectional one that is concluded and has provided the data used for the present work, and a prospective one that is planned to start. For the first phase of the study, the investigators recorded data on clinical, ECG, echocardiography, laboratory, and hemodynamic features, as well as on medical therapy, for all consenting consecutive ≥70 year-old patients with PAH or CTEPH evaluated between 1st December 2019 and 15th September 2020. The earliest visit performed during the study period was the reference, and missing information at the time of the reference visit was added if available within 3 months. Hemodynamic measurements at the time of the diagnosis were also retrieved, even if they were obtained before 70 years of age.

All patients were managed at each PH center in accordance with contemporary international guidelines. Data were entered into an electronic clinical report form (eCRF) using RedCap (Research Electronic Data Capture) [12]. The study protocol was in accordance with the ethical guidelines of the 1975 Declaration of Helsinki, and the registry was initially approved by the institutional ethics committee of the IRCCS Ospedale Policlinico San Martino in Genova (coordinating center, approval 421/2018 CER Liguria). 

For the present analysis, only individuals with a diagnosis of PAH made at 65 years of age or older were considered. Moreover, we excluded subjects without the following hemodynamic parameters available from the last RHC: mPAP, pulmonary artery wedge pressure (PAWP), cardiac output (CO), and PVR.

The included patients were divided into two groups according to the presence of elements suggestive for LHD, as established by two different approaches (Table 1).

For the main analysis, a LHD phenotype was defined as in the AMBITION trial [11]. As a secondary analysis, we expanded the clinical criteria suggestive for LHD, including permanent atrial fibrillation (AF) and echocardiographic parameters if the patients had only 2 risk factors for LV dysfunction (see Table 1 for details).

Normality was assessed with the Shapiro-Wilk test. Continuous variables are presented as mean ± standard deviation (SD) or median [interquartile range, IQR], depending on the distribution. Categorical variables are reported as absolute count and percentages. For comparison of normally distributed continuous data, group differences were tested by means of 2-sided student t test, while the 2-sided Mann-Whitney test was used for non-normally distributed variables. Frequency distributions between groups were compared with the chi-squared test or Fisher exact test, as appropriate.

*p*-values < 0.05 were considered as statistically significant. All statistical analyses were performed using R software (R version 3.6.1).

## 3. Results

One-hundred eighty elderly patients with PAH or CTEPH were included in the registry between 1 December 2019 and 15 September 2020 (Appendix A summarizes the contribution of each center). After excluding those who had been diagnosed with PAH before 65 years of age, with CTEPH, with post-capillary PH at diagnostic RHC, and/or without complete hemodynamic profile, 69 patients were included in the analysis (Figure 1, Table 2). These subjects were mostly female (64%) with a PAH diagnosis made at a mean age of 73 ± 4 years. At the first RHC, mPAP was 44 ± 12 mmHg, mean PAWP 10 ± 3 mmHg, mean cardiac index (CI) 2.1 ± 0.8 L/min, and mean PVR 9.1 ± 4.3 WU. Comorbidities were common: 45 (65%) had systemic hypertension, 17 (25%) had CAD, 31 (45%) had chronic obstructive pulmonary disease or interstitial lung disease (although of severity not deemed sufficient to account for group 3 PH), and 26 (38%) had CKD. The last available RHC was performed at a median of 15 (IQR 4–33) months after PAH diagnosis, and the hemodynamic profile was characterized by mean PAWP, PVR, and CI of 11 ± 4 mmHg, 7.39 ± 4.53 WU and 2.74 ± 0.81 L/min/m^2^, respectively. Sixty-three (91%) patients were taking PAH drugs and 36 (52%) were taking dual oral combination therapy.

According to the main analysis criteria, 23 (33%) patients had a LHD phenotype: 17 based on hemodynamic criteria and six based on clinical criteria (Figure 2). Of note, no patient with hemodynamic parameters suggestive for LHD had ≥3 risk factors for LV diastolic dysfunction, and the six patients with clinical criteria did not have pulmonary hemodynamics indicating LHD at the last RHC.

The characteristics of the patients with and without a LHD profile are listed in Table 2. Patients with a LHD phenotype more commonly had diabetes and CKD, and had higher PAWP and lower PVR. They also had higher right atrial pressure (RAP), although not to a significant extent, but a similar RAP/PAWP ratio. There were no differences in functional class, 6MWD, and echocardiographic parameters. No substantial disparities in PAH treatment were identified, while there was a greater use of renin-angiotensin inhibitors (RASi) and statins in patients meeting the LHD criteria (Table 2).

Patients’ characteristics according to whether they had clinical or hemodynamic criteria for a LHD phenotype are shown in Appendix A. Patients with a clinical LHD profile had a burden of comorbidities similar to the one in the primary analysis, but a similar rate of RASi use and lower PAWP than those without clinical LHD characteristics. Patients with a hemodynamic LHD phenotype had comparable prevalence of systemic hypertension, diabetes, and CAD compared with the non-LHD group, but higher RAP and lower peak tricuspid regurgitant velocity. They also had a greater use of RASi and amiodarone.

According to the secondary analysis criteria, 37 (54%) patients were classified as having a LHD phenotype (Figure 2): 20 based on the modified clinical criteria, 8 based on the hemodynamic criteria, and 9 patients based on both (Appendix A). As expected, subjects with a profile suggestive for LHD showed higher frequency of LV hypertrophy, left atrial dilation, and comorbidities. They also had higher RAP and non-significantly lower PVR. No differences in PAH therapy were observed, while patients with hemodynamics indicative of LHD were more frequently on RASi. 

Twenty-nine patients had a LHD phenotype using only the expanded clinical criteria (Appendix A). Of note, 16 (55%) of them had a PAWP < 13 mmHg at the last available RHC and 8 of the 40 subjects without a clinical LHD profile had hemodynamic criteria for LHD.

## 4. Discussion

The demographics of patients with PH have changed over time, with an increasing number of elderly individuals with PAH [3,4,5,6,7,8,9,13,14,15]. 

Here, we evaluated a real-world population of subjects with a diagnosis of PAH made after 65 years of age. The cohort we assessed had a median age at PAH diagnosis of 73 years, in line with the recent literature. In the COMPERA registry, the median age of incident idiopathic PAH was 71 years, and as many as 63% of the patients were older than 65 years [8]. We found a high rate of comorbidities, such as systemic hypertension (65%), diabetes (22%), and CAD (25%), again similar to that observed in previous investigations. In the subgroup of patients who were at least 75 years enrolled in the Swedish SPAHR registry, the prevalence of these conditions was 66%, 30%, and 26%, respectively [10]. 

Thus, we confirm that comorbidities that predispose to LHD are common among old subjects with PAH. This clinical background sheds doubts about the reliability of PAH diagnosis when PAWP is just slightly below or exactly at the 15-mmHg threshold. Following this reasoning, it is now recommended to identify patients with a LHD phenotype by combining clinical, ECG, and echocardiographic information, instead of relying only on the value of PAWP [1,16]. Indeed, RHC can fail to discriminate between pre- and post-capillary PH [1,17,18].

According to the criteria applied in the AMBITION trial, 33% of patients in our cohort had a LHD phenotype. This proportion rose to 54% when other parameters suggestive of LHD were taken into account. It can be argued that many of these cases might have been misclassified as pre-capillary PH at the first RHC. However, we included only patients with hemodynamic measurements consistent with the definition of pre-capillary PH. In addition, the patients enrolled in the PATRIARCA registry were followed at dedicated PH centers, and PAWP at the last available RHC was 11 ± 4 mmHg, similar to that reported in the COMPERA (10 ± 3 mmHg) and REVEAL (9 ± 4 mmHg) registries [6,8]. Interestingly, our primary analysis highlighted a disconnection between clinical and hemodynamic criteria to suspect occult LHD, which we believe is worth being further investigated.

The implications of the coexistence of PAH and a LHD phenotype are important: clinical outcomes tend to be worse [9,10,19], and pulmonary vasodilators could be detrimental [1,20,21].

Recently, McLaughlin et al. compared the patients with a LHD phenotype recruited during the first months of the AMBITION trial (ex-primary analysis set) and those enrolled after the adoption of a protocol amendment aiming at reducing the risk of subclinical LHD (primary analysis set) [11]. The ex-primary analysis set showed benefit from PAH treatment, but this was less pronounced as compared with the primary analysis set. Furthermore, they had a greater incidence of adverse events and study drug discontinuation [11].

Most subjects in PATRIARCA were treated with pulmonary vasodilators, with half taking dual oral combination therapy. This frequency is higher than the ones reported in COMPERA (31.6% one year after PAH diagnosis) and in the Swedish registry (14% and 9% in the age groups 65–74 years and ≥75 years, respectively) [8,10]. This finding could partly depend on the different periods covered previously and by our studies, since sequential or upfront use of more than one drug was optional in the past, but has recently become the standard of care [1,22,23]. Nonetheless, only 9% of the sample we analyzed was treated with triple oral therapy, including selexipag. No patient received subcutaneous treprostinil, and intravenous epoprostenol was administered only to one subject, indicating that treatment of PAH in the elderly is less aggressive than recommended, as already described [24].

The latest clinical practice guidelines form the European Society of Cardiology and the European Respiratory Society put emphasis on the increasing number of patients with idiopathic PAH and cardiopulmonary comorbidities, recommending initial monotherapy with an endothelin receptor antagonist or a phosphodiesterase type 5 inhibitor, irrespective of risk stratification, given the higher risk of fluid retention in this population [25]. Yet, a retrospective analysis of PAH patients treated at Amsterdam UMC showed an improvement in hemodynamic and imaging parameters despite the presence of a LHD phenotype, as identified by a high H2FPEF score [26]. About 90% of these subjects received pulmonary vasodilators and around 40% received double oral combination therapy. It is noteworthy that changes in the 4-strata risk profile were comparable among patients with low- and high-H2FPEF scores [26]. We do not have longitudinal data to determine the effects of PAH drugs in the PATRIARCA cohort. However, the large use of these medications suggests that they were well-tolerated.

This study is limited by the observational nature, with the possibility of selection and survival bias, lack of standardization of registered variables, and missing follow-up data. Nevertheless, the participating centers enrolled all consecutive patients meeting the inclusion criteria. The sample size was modest because we focused on a subgroup of subjects with a rare disease and who were enrolled during the SARS-CoV-2 pandemic. The small number of included patients is the most likely explanation for the lack of statistically significant difference in the prevalence of some risk factors for LHD, such as hypertension and CAD, in the distribution of WHO-FC classes, and in 6MWD between the no-LHD and LHD groups. We also acknowledge that other scores and approaches may be employed to predict occult LHD in individuals with PAH. Finally, echocardiographic and hemodynamic parameters were obtained from the last available examination, with a possible influence of the therapies prescribed meanwhile.

## 5. Conclusions

In the real world, a substantial proportion of elderly PAH patients is treated with pulmonary vasodilators despite having clinical or hemodynamic clues of LHD. Clinical trials and large observational studies are needed to better evaluate PAH drugs in this population.

## Figures and Tables

**Figure 1 jcm-11-07136-f001:**
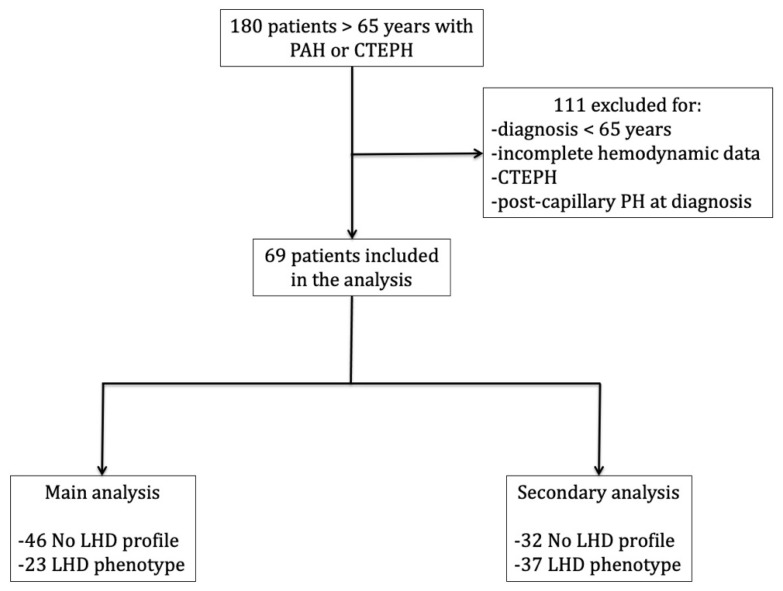
Flow-chart depicting the selection process of the study sample. *PAH*, pulmonary arterial hypertension; *CTEPH*, chronic thromboembolic pulmonary hypertension; *PH*, pulmonary hypertension; *LHD*, Left heart disease.

**Figure 2 jcm-11-07136-f002:**
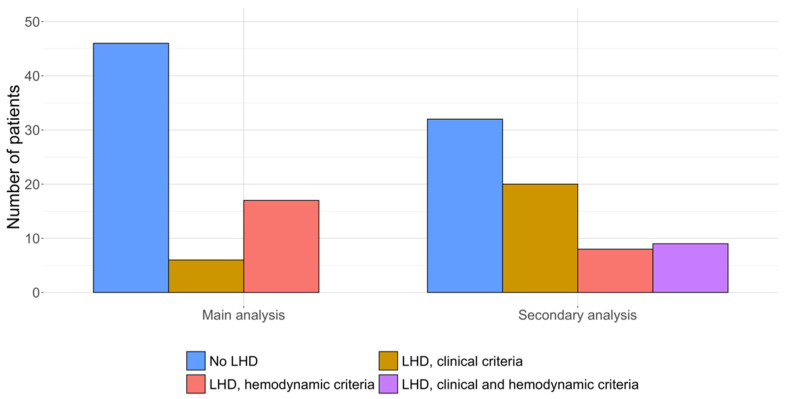
Number of patients with and without a left heart disease (LHD) phenotype according to main and secondary analysis criteria.

**Table 1 jcm-11-07136-t001:** Criteria for left heart disease phenotype definition.

Main Analysis
Patients with at least one of the following criteria:
(i) Clinical criteria: ≥ 3 of the following risk factors for LV diastolic dysfunction
-BMI ≥ 30 Kg/m^2^
-history of systemic hypertension
-diabetes mellitus (any type)
-history of significant CAD
(ii) Hemodynamic criteria:
-PVR between 3 and 3.75 WU
-PVR between 3.75 and 6.25 WU in the presence of PAWP between 13 and 15 mmHg
**Secondary analysis**
Patients with at least one of the following criteria:
(i) Clinical criteria. Either:
(ia) ≥3 of the following risk factors for LV diastolic dysfunction
-BMI ≥ 30 Kg/m^2^
-history of systemic hypertension
-diabetes mellitus (any type)
-history of significant CAD
(ib) 2 risk factors for LV diastolic dysfunction and ≥1 of the following:
-permanent AF
-LV hypertrophy
-LVEF <50%
-at least moderate mitral or aortic valve disease
-LA dilation
(ii) Hemodynamic criteria:
-PVR between 3 and 3.75 WU
-PVR between 3.75 and 6.25 WU in the presence of a PAWP between 13 and 15 mmHg

Clinical and hemodynamic criteria for left heart disease definition for the main and secondary analysis. *AF*, atrial fibrillation; *LV*, left ventricular; *BMI*, body mass index; *CAD*, coronary artery disease; *LVEF*, left ventricular ejection fraction; *LA*, left atrium; *PVR*, pulmonary vascular resistance; *PAWP*, pulmonary artery wedge pressure.

**Table 2 jcm-11-07136-t002:** Characteristics of the study population according to the main analysis criteria.

Main Analysis	Overall (*n* = 69)	No LHD Phenotype (*n* = 46)	LHD Phenotype (*n* = 23)	*p*
Demographics				
Age at diagnosis, years	73 ± 4	73 ± 4	73 ± 4	0.95
Age at enrolment, years	77 ± 5	77 ± 4	77 ± 4	0.77
Female	44 (64)	29 (63)	15 (65)	0.86
Weight, Kg	64 ± 15	63 ± 14	69 ± 16	0.13
Height, cm	163 ± 9	162 ± 9	165 ± 8	0.20
BSA, m^2^	1.70 ± 0.22	1.67 ± 0.22	1.76 ± 0.22	0.11
BMI, Kg/m^2^	24 ± 5	24 ± 4	25 ± 5	0.27
Clinical and echocardiographic parameters				
WHO-FC I-II	32 (46)	21 (46)	11 (48)	0.80
Systemic hypertension	45 (65)	27 (59)	18 (78)	0.11
Diabetes	15 (22)	6 (13)	9 (39)	**0.01**
CAD	17 (25)	8 (17)	9 (39)	0.05
Permanent AF	3 (4)	2 (4)	1 (4)	1
Pulmonary disease	31 (45)	22 (48)	9 (39)	0.49
CKD	26 (38)	12 (26)	14 (61)	**0.003**
SBP, mmHg	124 ± 15	124 ± 15	124 ± 15	0.98
DBP, mmHg	72 ± 9	73 ± 9	70 ± 9	0.22
SO_2_, %	95 [93; 97]	95 [93; 97]	95 [93; 97]	0.76
6MWD, meters	304 ± 199	315 ± 116	278 ± 127	0.35
LVEF <50%	4 (6)	2 (4)	2 (9)	0.22
LVH	17 (25)	10 (22)	7 (30)	0.37
LA dilation	31 (45)	19 (41)	12 (52)	0.48
TAPSE, mm	20 ± 5	21 ± 4	19 ± 5	0.27
TRV, m/s	3.81 ± 0.76	3.91 ± 0.71	3.60 ± 0.82	0.12
TAPSE/TRV	5.6 ± 1.9	5.5 ± 1.9	5.8 ± 2.1	0.68
RVSP, mmHg	57 [42; 77]	62 [43; 80]	48 [38; 66]	0.15
RA dilation	53 (77)	34 (74)	19 (83)	0.38
Pericardial effusion	11 (16)	8 (17)	3 (13)	0.64
Most recent RHC				
RAP, mmHg	7 [4; 10]	6 [3; 9]	8 [5; 11]	0.06
mPAP, mmHg	41 ± 10	42 ± 11	38 ± 8	0.10
dPAP, mmHg	25 ± 9	27 ± 10	23 ± 6	0.13
sPAP, mmHg	70 ± 20	71 ± 21	66 ± 18	0.31
PAWP, mmHg	11 ± 4	10 ± 3	14 ± 5	**<0.001**
RAP/PAWP ratio	0.60 [0.46; 0.75]	0.60 [0.40; 0.75]	0.67 [0.46; 0.81]	0.61
PVR, WU	7.39 ± 4.53	8.30 ± 4.80	5.56 ± 3.31	**0.02**
Cardiac output, L/min	4.59 ± 1.43	4.46 ± 1.55	4.85 ± 1.15	0.29
Cardiac index, L/min/m^2^	2.74 ± 0.81	2.67 ± 0.82	2.88 ± 0.79	0.30
Diagnosis to last RHC interval, months	15 [4; 33]	13 [4; 30]	18 [5; 38]	0.59
Treatment				
No PAH therapy	6 (9)	4 (9)	2 (9)	1
Bosentan	-	-	-	
Ambrisentan	17 (25)	12 (26)	5 (22)	0.69
Macitentan	32 (46)	21 (46)	11 (48)	0.74
ERA	49 (71)	33 (72)	16 (70)	0.85
Sildenafil	22 (32)	11 (24)	11 (48)	0.05
Tadalafil	25 (36)	18 (39)	7 (30)	0.48
Riociguat	3 (4)	3 (7)	0	0.21
PDE5i/GCs	50 (73)	32 (70)	18 (78)	0.45
Dual oral combination therapy	36 (52)	23 (50)	13 (57)	0.61
Selexipag	6 (9)	4 (9)	2 (9)	1
Treprostinil	-	-	-	
Epoprostenol i.v.	1 (1)	1 (2)	0	0.47
Inhaled iloprost	2 (3)	2 (4)	0	0.31
Beta blockers	9 (13)	5 (11)	4 (17)	0.45
RASi	23 (33)	10 (22)	13 (57)	**0.004**
MRA	32 (46)	20 (44)	12 (52)	0.44
Furosemide	57 (83)	38 (83)	19 (83)	1
Digoxin	6 (9)	5 (11)	1 (4)	0.37
Amiodarone	9 (13)	4 (9)	5 (22)	0.14
Warfarin	11 (16)	8 (17)	3 (13)	0.69
DOAC	11 (16)	6 (13)	5 (22)	0.31
SAPT	22 (32)	12 (26)	10 (44)	0.14
Statins	26 (38)	13 (28)	13 (57)	**0.02**
Ezetimibe	3 (4)	1 (2)	2 (9)	0.22
Antidiabetic drugs	12 (17)	4 (9)	8 (35)	**0.007**

Characteristics of patients with and without a left heart disease (LHD) phenotype according to the main analysis criteria. Data are expressed as *n* (%), mean ± SD, or median [IQR], as appropriate. *BSA*, body surface area; *BMI*, body mass index; *PAH*, pulmonary arterial hypertension; *WHO-FC*, World Health Organization functional class; *CAD*, coronary artery disease; *AF*, atrial fibrillation; *CKD*, chronic kidney disease; *SBP*, systolic blood pressure; DBP, diastolic blood pressure; *SO_2_*, oxygen saturation; *6MWD*, six minute walking distance; *LVEF*, left ventricular ejection fraction; *LVH*, left ventricular hypertrophy; *LA*, left atrium; *TAPSE*, tricuspid annular plane systolic excursion; *TRV*, tricuspid regurgitant velocity; *RVSP*, right ventricular systolic pressure; *RA*, right atrium; *RHC*, right heart catheterization; *RAP*, right atrial pressure; *mPAP*, *dPAP* and *sPAP* for mean, diastolic and systolic pulmonary artery pressure; *PAWP*, pulmonary artery wedge pressure; *PVR*, pulmonary vascular resistance; *ERA*, endothelin receptor antagonist; *PDE5i*, phosphodiesterase type 5 inhibitor; *GCs*, guanylate cyclase stimulator; *RASi*, renin-angiotensin system inhibitors; *MRA*, mineralcorticoid receptor antagonist; *DOAC*, direct oral anticoagulant; *SAPT*, single antiplatelet therapy.

## Data Availability

The data used for the analysis presented in the manuscript can be provided by the corresponding author upon request.

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
