# Peer review of "Left Heart Disease Phenotype in Elderly Patients with Pulmonary Arterial Hypertension: Insights from the Italian PATRIARCA Registry"

_jcm, 2022, doi:10.3390/jcm11237136_

Round 1

Reviewer 1 Report

Dear editor,

Thank you for the opportunity to review the paper focused on the impact of LV phenotype on RV and pulmonary haemodynamic profile and clinical management in elderly (≥70 years) patients with pulmonary arterial hypertension (PAH).

The topic of the paper is highly clinically relevant and high-quality data are pulled from the national registry.

The main conclusion of the paper, which should be taken seriously, is that in the real world, elderly PAH patients were treated with pulmonary vasodilators, including combinations, despite a remarkable prevalence of an LHD phenotype.  The high prevalence of the coexistence of PAH and LHD phenotype begets worse clinical outcomes per se. Furthermore, this deadly duo frequently brings a heavy dilemma in front of the clinicians regarding the treatment strategy, since pulmonary vasodilators could in principle be detrimental in this cohort of patients.

This study raises a quest for new clinical trials and large observational studies evaluating PAH drugs in this population. It might be that this population is heterogeneous, encompassing several different endotypes that would need different management strategies based on the results of the standardized hemodynamic and humoral testing.      

Author Response

We thank the Reviewer for the positive comment on our work

Reviewer 2 Report

First I would like to congratulate authors for presenting this manuscript on a difficult subject. But I have some reservations on the way the study was conducted

1) Even though there are 2 different study populations, there is only minor difference between their clinical profile. Only Diabetes and CKD are statistically significant in LHD Phenotype group. Hypertension, CAD and Obesity have similar incidence in both groups.

2) Non LHD group have LV Hypertrophy , LA Dialatation and LV dysfunction . Then how did they do not come in LHD group ?

3)Unlike prior studies, there is no difference between clinical class and 6 minute walk tests between both groups. How do the authors substantiate this ?

4) I fully agree with authors that there is disconnection between clinical and hemodynamic criteria to suspect occult LHD. This has made both groups nearly identical.

5) I strongly feel the study has major limitations like significant selection bias, missing follow up data , Echo/ hemodynamic parameters at initial presentation etc. I feel this has significantly affected study results.

I strongly feel the authors should initiate a clinical trial evaluating PAH drugs and prognosis in elderly PAH patients diagnosed and followed by same  PAH clinics which will be interesting.

Author Response

We thank the Reviewer for the positive comment on our work.

We agree that the lack of statistically significant difference in risk factors for LHD between the study groups, as well as in the distribution of NYHA classes and 6MWD,  is surprising. We hypothesize that this is due to the small sample size - indeed, these variables were in the expected direction although statistical significance was not reached. This is another argument supporting the call for additional and larger studies on the topic.

We also acknowledge that other approaches can be used to predict the presence of occult LHD in PAH, including the simple assessment of LV hypertrophy or dysfunction by echocardiography.

Finally, it is possible that our study is flawed by the biases typical of retrospective investigations (although the participating centers enrolled all consecutive patients meeting the inclusion criteria).

All these limitations are clearly discussed at the end of page 9 and the beginning of page 10 of the revised manuscript, which has also been revised for English language.

Reviewer 3 Report

The manuscript entitled " Left heart disease phenotype in elderly patients with pulmonary arterial hypertension: insights from the Italian PATRIARCA registry" evaluated the presence of a LHD phenotype and the pattern of pulmonary vasodilator prescription in a cohort of elderly patients with PAH enrolled in the multicenter PATRIARCA registry.  It is an interesting topic and provide important real world infromation. However, the part of "materail & method" was a little too long and should be revised.    

Author Response

The Methods section has been shortened as suggested.

Furthermore, the text has been revised for English language